# A Dynamically Dimensioned Search Allowing a Flexible Search Range and Its Application to Optimize Discrete Hedging Rule Curves

**Youngkyu Jin [1]**, **Sangho Lee [2]**, **Taeuk Kang [3]** and **Yeulwoo Kim [2,\*]**

1  Industry-University Cooperation Foundation, Pukyong National University, Busan 48513, Korea
2  Civil Engineering, Department of Sustainable Engineering, Pukyong National University, Busan 48513, Korea
3  Disaster Prevention Research Institute, Pukyong National University, Busan 48513, Korea
\*  Correspondence: yarkim@pknu.ac.kr; Tel.: +82-51-629-6076

**Abstract:** The discrete hedging rule for reservoir operation includes time-varying trigger volumes used for the onset and termination of water rationing, which complicates its optimization problems. A dynamically dimensioned search can be easily applied to complex optimization problems, but the performance is relatively limited in constrained optimization problems such as deriving reservoir operation rules. A dynamically dimensioned search allowing for a flexible search range is proposed in this study to efficiently solve constrained optimization problems. The modified algorithm can recursively update the search ranges of decision variables with limited overlaps. The above two algorithms are applied to derive hedging rule curves for three reservoirs. Objective function values are closely converged to optimum solutions, with fewer evaluations using the modified algorithm than those using the traditional algorithm. The modified algorithm restrains an overlapped search range of decision variables and can reduce redundant computational efforts caused by unreasonable candidate solutions that violate inequality conditions.

**Keywords:** optimization; constraint; dynamically dimensioned search; hedging rule; reservoir



## 1. Introduction

A purpose of a reservoir is to regulate natural river flow fluctuations by storing the excess water during the wet period, which is then released during the dry period to meet municipal, agricultural, and instream flow [1,2]. Severe and prolonged droughts, however, may lead to reservoir releases that are insufficient to satisfy the planned water supply. As one of the reservoir operation rules to cope with droughts, the hedging rule promotes minor shortages in advance of deficits to reduce the probability of emptying the reservoir, and consequently of severe impacts [3–5].

Researchers have generally derived hedging rules for reservoir operation from optimization models, such as linear, nonlinear, and dynamic programming [6–13]. Recently, several studies have proposed methods to derive optimal hedging rules using evolutionary or heuristic algorithms such as genetic algorithms, particle swarm optimization, and dynamically dimensioned search algorithms [14–18]. The derivation of optimum hedging rules must consider many factors to cope with droughts efficiently, e.g., the percentage of water rationing and trigger volume (onset and termination of hedging) varying with time or phase. Many factors make the optimization problem difficult by increasing the number of decision variables and constraints. Thus, a proper optimization method and strategy are essential in complex optimization problems utilizing/generating hedging rules.

In some previous researches, the typical decision variables were trigger volumes in the optimization problem for the hedging rules [19–22]. Shih and Revelle [23] suggested a discrete hedging rule consisting of several rationing phases. They then implemented mixed-integer linear programming to seek the monthly trigger volumes of each rationing phase.



In their optimization model to determine the optimal discrete hedging rule, they used many constraint equations to define and to separate the zones, and assigned the supply quantity at each zone. For this reason, some subsequent studies focused on the discrete hedging rule applying linear programming or nonlinear programming [7,13,15,23]. It is often impractical or unsuitable to apply linear and nonlinear programming to solve constrained optimization problems because the amount of computation required becomes unmanageable as the problem size increases; the constraints violate the required assumptions [24]. Furthermore, a reservoir operation model based on the discrete hedging rule requires many prescribed decision variables. Thus, linear programming requires excessive computational time when solving large-sized optimization problems [25]. Dynamic programming also requires significant computational time in large-sized optimization problems with constraints.

Some researchers have applied heuristic methods to solve optimization problems in the water resources field due to the flexibility and efficiency in searching for optimum solutions [26–29]. Usage of the heuristic methods may alleviate the above difficulties. Recently, researchers have suggested various heuristic algorithms to emulate several natural phenomena or to use stochastic algorithms to solve complex optimization problems [30–32]. They are the genetic algorithm (GA), the Shuffled Complex Evolution-University of Arizona (SCE-UA), and the dynamically dimensioned search algorithm (DDS).

Schematic diagrams of the GA, SCE-UA, and DDS algorithms are presented in Figure 1. GA and SCE-UA are meta-heuristic methods that belong to the class of evolutionary algorithms. GA searches for the optimal solution using biological operators such as mutation, crossover, and selection (Figure 1a). GA has two major parameters that influence the determination of candidate solutions: mutation probability and crossover probability. If the mutation probability is too small, the candidate solution groups are generated only by the combination of the initial solution groups. That is, with the too-small mutation probability, it may be difficult to search for a good solution. When the mutation probability is too high, it may cause a loss of a good solution. Therefore, GA requires tuning to determine an appropriate combination of parameters. More details about GA can be found in [30,33–35]. The SCE-UA employs a process called competitive complex evolution (CCE) algorithm to search for a global optimization (Figure 1b). The SCE generates an initial solution within the feasible space of the parameter and divides it into complexes. Each complex evolves using a downhill simplex algorithm (reflection and contraction) [36]. The evolved complexes are shuffled. The process of evolution and shuffle is repeated until the convergence condition is satisfied [31,34,37,38]. However, Chu et al. [39] suggests that when SCE-UA was designed, SCE-UA was constructed primarily for and tested on low-dimensioned problems. Chu et al. [39] reveals that SCE-UA tends to malfunction on high-dimensional problems, due to the fact that shuffled points may converge within a subspace of the original search space [40,41]. The DDS is a single-solution-based heuristic neighborhood search algorithm and has been produced by Tolson and Shoemaker [32]. Unlike evolutionary algorithms, the DDS is designed to find good global solutions within a specified maximum number of function evaluations. The DDS globally searches for candidate solutions early in the exploration, and the search dimension gradually decreases as the number of function iterations increases. The candidate solution is determined by perturbing the current best solution in the randomly sampled dimensions only [42].

Kang et al. [43] compared the three heuristic methods (i.e., the DDS, SCE-UA, and GA) from their ability to search for reasonable solutions close to the global minima of the six test functions: the Bukin, Schubert, Michalewicz, Griewank, Rastrigin, and Schwefel functions. The evaluation results showed that the DDS was better than SCE-UA and GA for the overall performance. According to the results of Kang et al. [43], as mentioned by Chu et al. [39], the performance of SCE-UA tended to underperform as the decision variables of the test functions increased. The GA also showed underperformance as the number of decision variables increased, similar to the SCE-UA. Then, they used the DDS to determine the 36 unknown trigger volumes, consisting of three hedging phases varying monthly, for the zone-based operation of a reservoir.

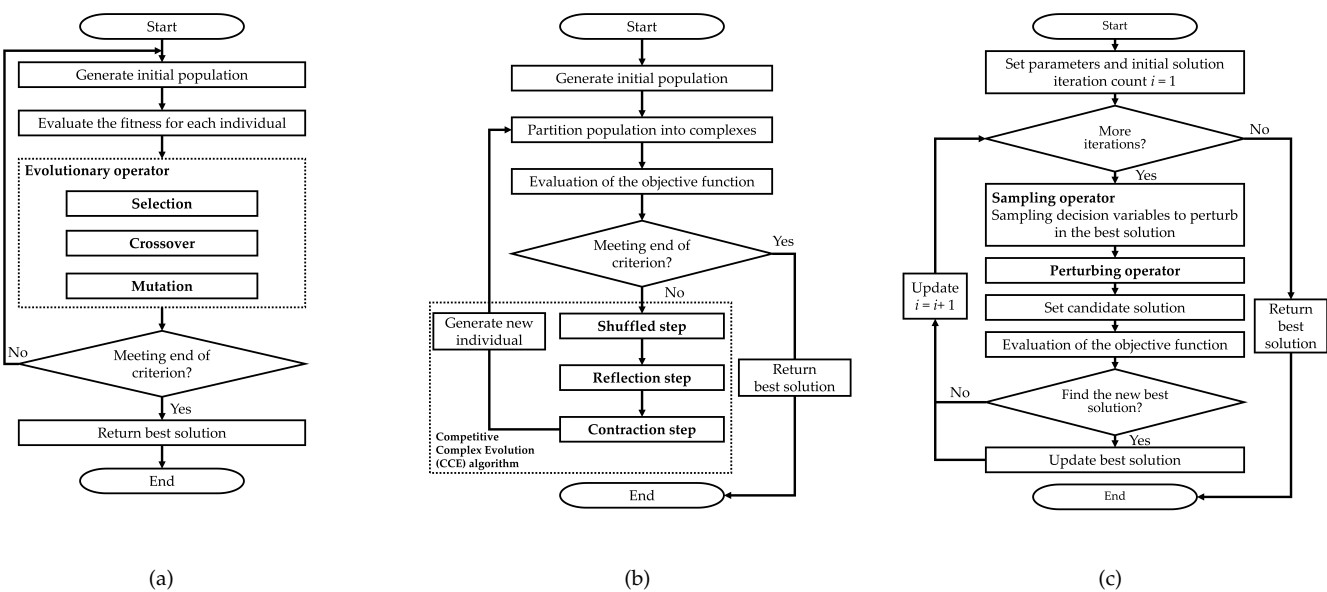

**Figure 1.** Schematic diagram of the three heuristic algorithms. (**a**) Genetic algorithm; (**b**) Shuffled Complex Evolution-University Arizona; (**c**) Dynamically dimensioned search algorithm.

In the discrete hedging rules, the trigger volume decreases from a minor rationing phase to a severe rationing phase, which should constrain the optimization problems. The traditional heuristic methods, however, are practically unconstrained optimization algorithms, limiting only the searching range of the decision variable. If heuristic methods are applied to unconstrained optimization for discrete hedging rules, many calculations are redundant due to unreasonable solutions in the search for the optimal solutions. Therefore, these algorithms must be used with additional mechanisms to implement constraints when solving constrained optimization problems [44]. Many alternative approaches have been introduced to solve constrained optimization problems, such as penalty functions, multiple-objective optimization techniques, and hybrid methods combining mathematical programming and evolutionary processes [45–47].

The purpose of the study is to improve the DDS algorithm to include constraints, and to apply the algorithm to an optimization of the discrete hedging rule. The DDS can be easily applied to complex optimization problems, but the performance is relatively inferior in constrained optimization problems because the search range of one decision variable may overlap with the search range of another. The overlapped search ranges may lead to the determination of unreasonable candidate solutions. Since the DDS terminates the optimization based on the maximum number of function evaluations, the unreasonable candidate solutions may cause a loss to a number of iterations. This study is to supplement the limitations of the DDS identified in the previous studies. Since it is not easy to derive a reasonable solution by implementing a one-time optimization process, Jin and Lee [48,49] derived the hedging rule curves using the DDS via a repeating optimization strategy. This strategy uses the solution derived from the previous optimization as the initial value of the subsequently attempted optimization. Chu et al. [40] noted that optimization algorithms developed using benchmark or random number functions to solve low-dimensional optimization problems have difficulties in high-dimensional problems. Furthermore, when problem dimensionality increases, these algorithms lose their effectiveness because the power of randomization drops geometrically [50–52]. Finally, Chu et al. [40] noted that algorithm developers should focus on solving problems instead of intricate benchmark functions.

The study here proposes a dynamically dimensioned search allowing a flexible search

range (DDS-FSR) to efficiently solve constrained optimization problems that can be specified by the problems for deriving hedging rule curves in the reservoir. In the optimization problem for the reservoir operation rule curve, the constraint is that the trigger volume of the moderate drought phase must be greater than the severe drought phase inspired the concept of the DDS-FSR. The DDS-FSR is modified based on the DDS. The difference between the two algorithms is as follows:

- The DDS has constant specified-search boundaries of decision variables until the optimization is terminated.
- If the two decision variables are under an inequality constraint, the DDS-FSR can exclude infeasible areas for the decision variable by converting the upper or lower boundary for the decision variable to the current best solution for the other decision variable.

## 2. Methods

### 2.1. Discrete Hedging Rules

Lund [53] describes various reservoir operation rules: standard operation rule, hedging rule, pack rule, and zone-based operation rule. The hedging rule can effectively manage water supply to cope with droughts. The hedging rule rations water supply below target levels and can lessen future water shortages. Shih and Revelle [23] presented a discrete hedging rule that uses the trigger volume to ration the release of each hedging phase. The optimization model for the discrete hedging rule decides the trigger volumes, which might vary monthly for the different rationing phases. The Korean drought contingency plan comprises four stages: concern, caution, alert, and severe. The discrete hedging rule's rationing phases need to correspond to the above four stages. The discrete hedging rule, including the four rationing phrases used in the research, is shown in Figure 2 [13].

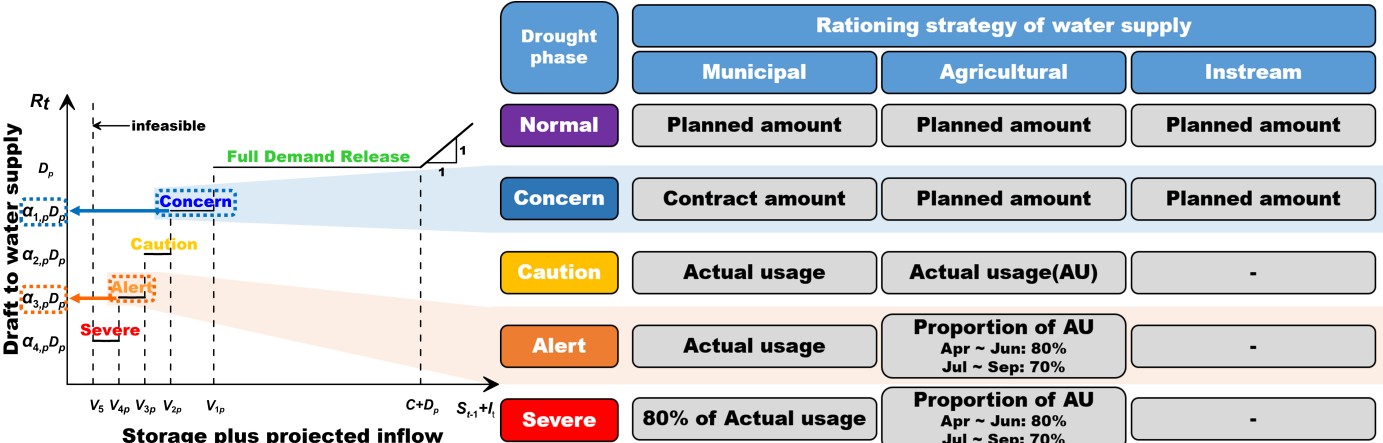

**Figure 2.** The discrete hedging rule with the water rationing strategy of the Korean drought contingency plan. The left figure is the schematic diagram of the discrete hedging rule, and the right figure is the water rationing strategy in the Korean drought contingency plan.

In Figure 2, $V_{1,p}$ is the trigger volume for concern phase above which no restrictions on water use are placed. $V_{2,p}$, $V_{3,p}$, and $V_{4,p}$ are the trigger volumes below which caution, alert, or severe phase are implemented, respectively, for month $p$. $V_5$ is the storage of low-water level for a reservoir. $D_p$ is the total planned water supply for month $p$. $\alpha_{1,p}$, $\alpha_{2,p}$, $\alpha_{3,p}$, and $\alpha_{4,p}$ are rationing factors for different drought phases in all month $p$. The rationing factor is a ratio of release corresponding to the drought phase and the total planned water supply. $S_{t-1}$ is the storage at the end of last period $t-1$, and $I_t$ is the current period inflow. The available water can be defined as $S_{t-1}$ plus $I_t$.' When, for example, the available water is on caution phase, the release $R_t$ equals $\alpha_{2,p}D_p$. In the case of multi-purpose dams in the Republic of Korea, the total planned water supply is divided into municipal water,

agricultural water, and instream flow water, based on the usage purpose. As shown in Figure 2, the Korean drought contingency plan presents different water strategies for each usage purpose of water supply. For example, the municipal water supply for each drought phase is described as follows:

- Normal: Release the monthly planned municipal water supply;
- Caution: Release the monthly contracted water supply with local governments/industrial complexes, etc.;
- Alert: Release the monthly actual usage surveyed on last year basis among the contractual water supply;
- Severe: Release 80% of the monthly actual usage.

In the alert and severe phases, the agricultural water is supplied 80% of the actual usage from April to June, which is 10% more than from July to September, considering the initial vegetative period of crops (Figure 2).

The trigger volumes from the concern to the severe drought phases decrease and are mutually apart for the following reasons: the drought phase sequentially escalates, and the water managers need time for decision-making to cope with droughts. Programming can easily establish specific trigger volumes in the period, $p$, using the following constraint equations.

$$V_{1,p} - V_{2,p} > 0 \tag{1}$$

However, the traditional heuristic methods are poor at including the inequality constraints of Equation (1) because the search range of one decision variable may overlap with the search range of another. The following section introduces a simple strategy to narrow overlaps on search ranges of decision variables in the DDS algorithm, which makes it easier to satisfy Equation (1).

### 2.2. Dynamically Dimensioned Search Allowing a Flexible Search Range

As a simple method to effectively satisfy the above constraints, the DDS-FSR is suggested, which may narrow overlaps on the searching ranges of decision variables while searching for an optimum solution.

The DDS is a point-to-point stochastic-based heuristic global searching algorithm suggested by Tolson and Shoemaker [32] to estimate the parameters of watershed run-off models. The main feature of the DDS was motivated by past experience from the manual calibration of watershed runoff and reservoir simulation models. They developed the DDS to search for an optimum solution of decision variables close to the global optimum solution within the specified maximum number of function evaluations. In early evaluations, the algorithm searches globally, searching for all decision variables. As the number of iterations approaches the maximum number of function evaluations, the algorithm locally searches for some decision variables. The transition from global to local search is achieved by dynamically and stochastically reducing the number of dimensions perturbed in the neighborhood of the current best solution. The search process of the DDS consists of three operators: sampling operator, perturbing operator, and decision operator. The sampling operator, controlled by sampling criteria probability ($P$), selects some decision variables perturbed from all the decision variables. The sampling criteria probability is a monotonically decreasing function of the iteration number ($i$), and the maximum number of function evaluations ($m$), written by Equation (2). The sampling operator assigns a random number uniformly distributed between 0 and 1 to each decision variable in an iteration. The sampling operator will perturb the decision variable whose random number assigned is less than the sampling criteria probability.

$$P(i) = 1 - \ln(i) / \ln(m) \tag{2}$$

The perturbing operator perturbs the current best solution, as shown in Figure 3, to generate the candidate solution for a selected decision variable. For the other decision

variables not selected by the sampling operator, the current best solutions are the candidate solutions.

In Figure 3, the variable $x_j$ is the decision variable that is an element of the decision vector $X = [x_1, x_2, \cdots, x_j, \cdots, x_J]^T$ and the variable $x_j^{best}$ is the current best solution searched up to the current iterations. The values $x_j^{max}$ and $x_j^{min}$ are upper and lower bounds for decision variable $x_j$. The vectors $X^{max}$ and $X^{min}$ are the set of $x_j^{max}$ and $x_j^{min}$ defined with the same dimension as $X$. The parameter $r$ is the neighborhood perturbation size that Tolson and Shoemaker [32] have recommended by 0.2. The function $N(0,1)$ is a standard normal distribution function with a mean of zero and a standard deviation of one. In summary, the user-defined $X^{max}$, $X^{min}$, and $r$ significantly affect the determination of the search ranges of the candidate solutions.

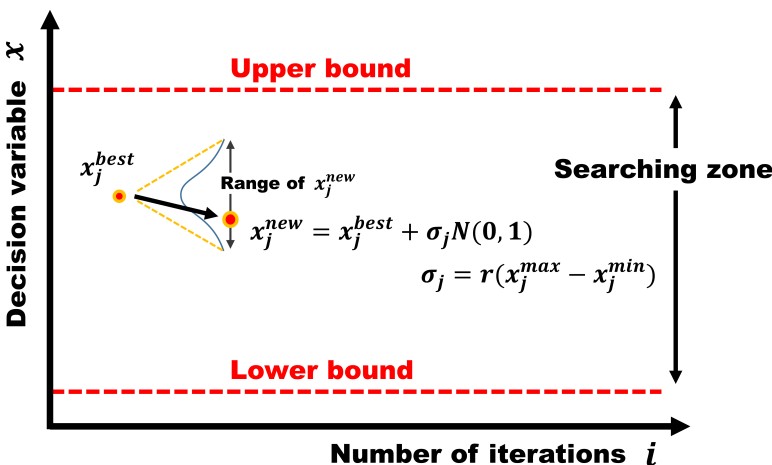

**Figure 3.** The schematic diagram of the perturbing operator in the DDS.

The decision operator serves to determine whether the candidate solution yields an objective function value inferior to the current best value. For example, when the objective function value of the candidate solution is superior to the current best value, the candidate solution is updated to the best solution.

The DDS was developed for unconstrained problems or bound-constrained problems. A simple way to optimize constrained problems using the DDS is the penalty methods, which seek the solution by replacing the original constrained problem with a sequence of unconstrained sub-problems, where the constraint functions are combined with the objective function to define a penalty function. In other words, the penalty method is to discard an infeasible solution by imposing a big number on the objective function if the candidate solution violates the constraints in the optimization problem seek to minima [54,55]. When solving the constrained optimization problem using the DDS, one can formulate the problem as follows:

$$Minimize \quad z = f(X) + \omega \sum_{j=2}^{J} CL_j \tag{3}$$

$$X = [x_1, x_2, \cdots, x_j, \cdots, x_J]^T \tag{4}$$

$$CL_j = \begin{cases} 1, & \text{if } x_{j-1} - x_j < 0 \\ 0, & \text{otherwise} \end{cases} \tag{5}$$

where $z$ is the objective function of the optimization problem with a penalty term; $X$ is the set of the decision variables; $\omega$ is the big number; and $CL_j$ is the binary variable that determines whether the candidate decision variables violate the inequality. Searching

for the optimal solution to the mentioned problem (Equations (3)–(5)) using the DDS is inefficient because the candidate solution might violate the inequality during the searching process (Figure 4a).

This study modified the perturbing operator as shown in Equation (6) to efficiently search for the optimal solution to the optimization problem with the inequality presented in Equations (3)–(5). Figure 4a shows all the decision variables have the same searching zone, and some ranges of candidate decision variables overlap, which yields unreasonable solutions. Figure 4b is the schematic diagram of a method to determine the search range of the DDS-FSR. Each decision variable has a reduced searching zone, and the ranges of candidate decision variables hardly overlap.

$$
\begin{aligned}
\sigma_1 &= r(x_1^{max} - x_2^{best}) \\
\sigma_j &= r(x_{j-1}^{best} - x_{j+1}^{best}) \quad, \quad for\ j = 2, \cdots, J-1 \\
\sigma_J &= r(x_{J-1}^{best} - x_J^{min})
\end{aligned}
\tag{6}
$$

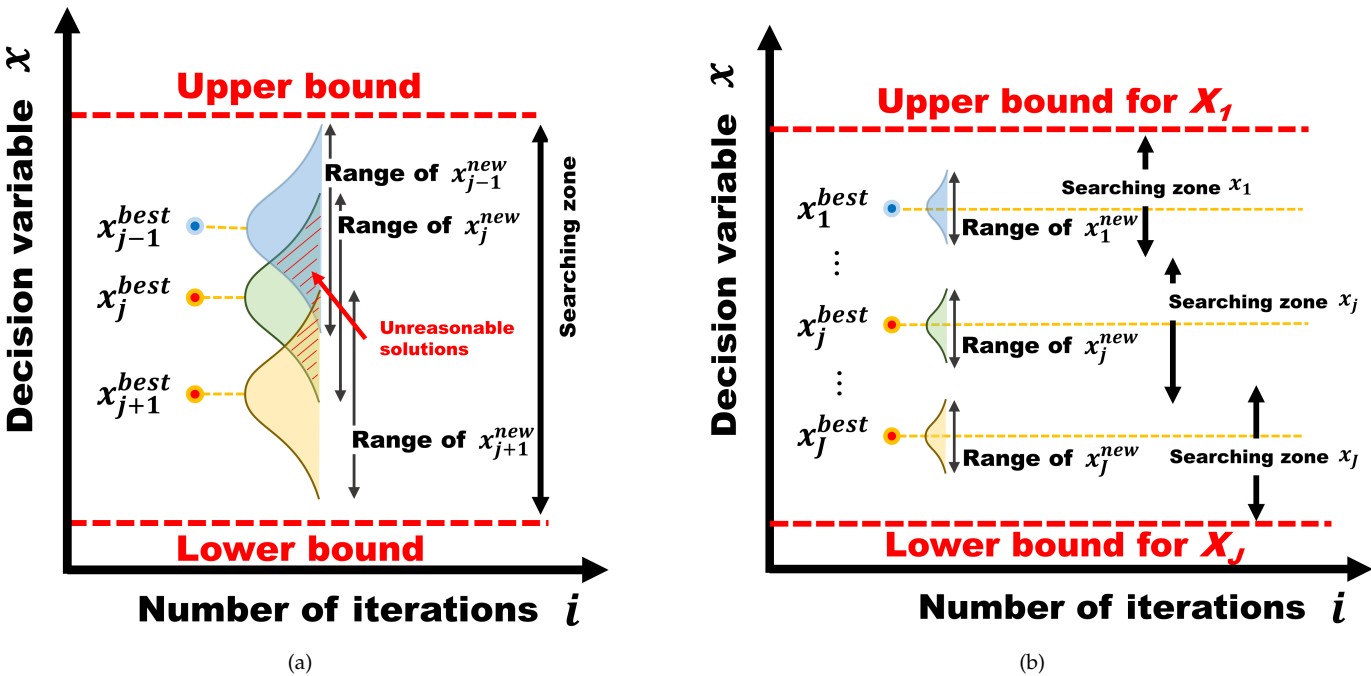

**Figure 4.** Comparison of search processes in the DDS and the DDS-FSR. (**a**) DDS; (**b**) DDS-FSR.

### 2.3. Reservoir Simulation and Optimization Model

The parameters of the monthly reservoir simulation model with hedging rule curves are monthly trigger volumes ($V_{1,p}$, $V_{2,p}$, $V_{3,p}$, $V_{4,p}$) and water rationing factors ($\alpha_{1,p}$, $\alpha_{2,p}$, $\alpha_{3,p}$, $\alpha_{4,p}$) in each drought phase for month $p$.

In the case of the Republic of Korea, the water rationing factor can be calculated according to the strategy of the water rationing (Figure 2). The water rationing factor is the ratio of release corresponding to the drought phase for the month to the total planned water supply, and can be calculated as follows. For example, the water rationing factor of the caution phase ($\alpha_{2,p}$) for the month $p$ can be formulated as follows:

$$
\alpha_{2,p} = \frac{AMD_p + AAD_p}{D_p}
\tag{7}
$$

where $AMD_p$ is the actual usage of municipal water supply for the month $p$, and $AAD_p$ is the actual usage of agricultural water supply for the month $p$. $D_p$ is the total planned water supply for the month $p$, and it is the sum of the planned municipal, agricultural, and instream flow water supply for the month $p$. Therefore, in this study, the optimization for

the parameters of the monthly reservoir simulation model was restricted to hedging rule curves. The hedging rule curves are bent lines that connect the monthly trigger volumes for each drought phase.

Optimizing the monthly trigger volumes for each drought phase with the DDS or the DDS-FSR requires formulation of the objective function and constraints equations. The objective function is the minimization of the three terms:

1. The sum of water supply shortage for the total period $T$;
2. The penalty term to restrain the reversal of trigger volumes in drought phase severity;
3. The penalty term to restrain water supply failures within the optimization period.

$$Minimize \quad z = \sum_{t=1}^{T} WS_t + \omega_1 \sum_{dp=2}^{4} \sum_{p=1}^{12} REV_{dp,p} + \omega_2 \sum_{t=1}^{T} Fail_t \tag{8}$$

$$WS_t = D_p - R_t \tag{9}$$

where $WS_t$ is the water supply shortage in period $t$. $D_p$ is the total planned water supply for the month $p$ corresponding to period $t$, and $R_t$ is the release in period $t$. $REV_{dp,p}$ is the binary variable that is a value of 1 or 0 related to the violation of the inequality, and $dp$ are the drought phases (1: concern, 2: caution, 3: alert, and 4: severe). For example, if $v_{2,p}$, the trigger capacity of the relatively serious drought phase, is greater than $v_{1,p}$, $REV_{2,p}$ becomes 1, and a big number is imposed on $z$. $\omega_1$ and $\omega_2$ are big numbers. $R_t$ is determined by the conditions of the water rationing for derived hedging rule curves. $Fail_t$ is a binary variable that is a 0 or 1 related to the release. When the water supply fails ($R_t$ equal 0), $Fail_t$ becomes 1, and a big number is imposed on $z$. After determining reservoir release, reservoir storage at the end of period $t$ can be calculated by following the water balance equation:

$$S_t = S_{t-1} + I_t - R_t - W_t \tag{10}$$

where $S_t$ is the storage at the end of period $t$, $I_t$ is the inflow in period $t$, and $W_t$ is the spill in period $t$. Table 1 shows the condition to classify the drought phase and their corresponding water rationing.

The parameters of the monthly reservoir simulation model with the hedging rule curves are monthly trigger volumes of the four drought phases. Thus, there are 48 decision variables (4 phases $\times$ 12 months) in the optimization problem for the hedging rule curves.

**Table 1.** Conditions and equations of the water rationing for reservoir simulation with the discrete hedging rule.

| Classification | Simulation Model | | |
| --- | --- | --- | --- |
| | Drought Phases | Condition | Release |
| Release determination | Normal | $S_{t-1} + I_t > V_{1,p}$ | $R_t = D_p$ |
| | Concern | $V_{2,p} < S_{t-1} + I_t \leq V_{1,p}$ | $R_t = \alpha_{1,p} D_p$ |
| | Caution | $V_{3,p} < S_{t-1} + I_t \leq V_{2,p}$ | $R_t = \alpha_{2,p} D_p$ |
| | Alert | $V_{4,p} < S_{t-1} + I_t \leq V_{3,p}$ | $R_t = \alpha_{3,p} D_p$ |
| | Severe | $V_5 < S_{t-1} + I_t \leq V_{4,p}$ | $R_t = \alpha_{,p4} D_p$ |
| | Fail | $S_{t-1} + I_t \leq V_{4,p}$ | $R_t = 0$ |

## 3. Study Area and Data

To compare the performance of the DDS and the DDS-FSR in the optimization problems with constraint equations, the two algorithms were applied to the optimization problems for hedging rule curves to three reservoirs with their own data, e.g., historic inflow, monthly planned water supply, and watershed area. The study cases are for the Andong (AD), Imha (IH), Hapcheon (HC), and Namgang (NG) reservoirs, which have records of failure to water supply due to droughts among the multi-purpose dams in the

Nakdong River basin in the Republic of Korea (Figure 5). A diversion pipeline connects the AD and IH reservoirs that are operated as a reservoir in practice. In this study, the AD and IH reservoirs were regarded as an equivalent reservoir, and one operation rule was derived.

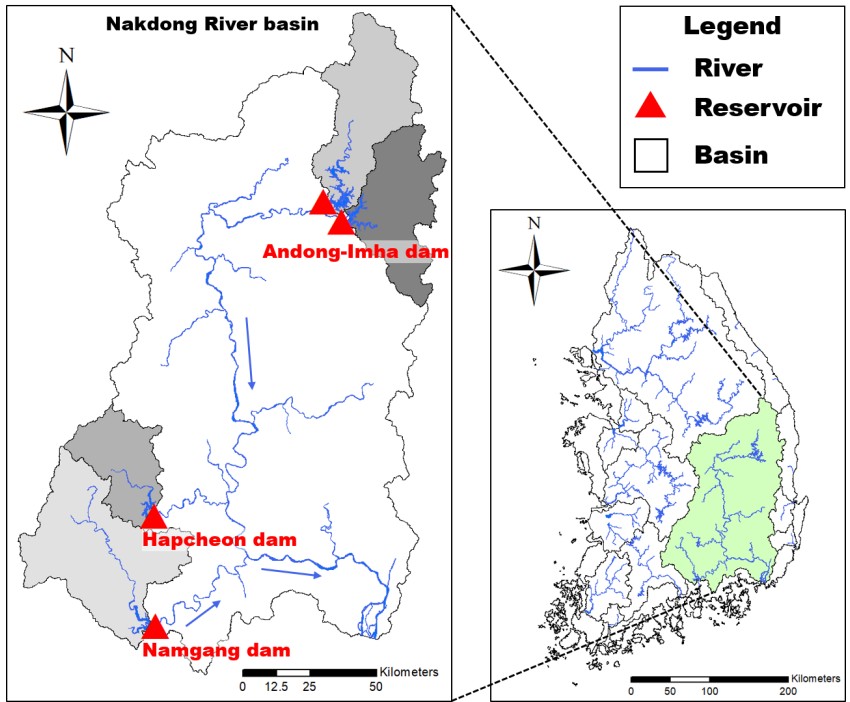

**Figure 5.** Location map of the adopted multi-purpose reservoirs.

The monthly planned water supply of each reservoir is presented in Figure 6. The water supply shown in Figure 6 is the monthly planned water supply of each dam designed at the time of construction. The water supply from the multi-purpose dam in the Rep. of Korea is divided into the municipal water supply, agricultural water supply, and instream flow, based on the purpose of the water supply. The planned water supply is calculated as a volume that satisfies about 97% of the temporal reliability from the water balance analysis for more than 20 years, considering the current and future demand. The temporal reliability ($rel_T$) is defined as the probability that the water supply is in a satisfactory state. *Rel* can be written as follows:

$$rel_T = \frac{n_s}{T} \times 100\% \tag{11}$$

where $n_S$ is the number of periods (month) during which demand is fully met, and $T$ is the total number of periods considered. The water demand is estimated using statistical data in the economic, humanities, and social fields in the past and present, and using indicators of mid-term financial plans by the government and local governments. In other words, the municipal water demand is estimated using the population, daily water usage per person, industrial complex site area, and daily water usage by industry in the water supply area, and agricultural water demand is estimated based on data from surveys on farmland, fields and paddies areas, etc. Instream flow demand is estimated considering the following items: the necessary flow rate to preserve water quality and aquatic ecosystems, to prevent saltwater intrusion, to protect water intake sources, and to maintain groundwater levels.

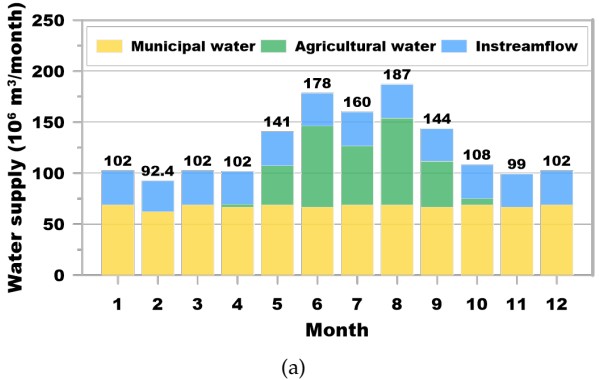

(a)

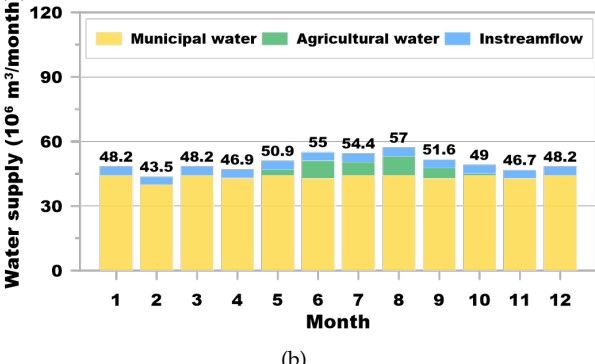

(b)

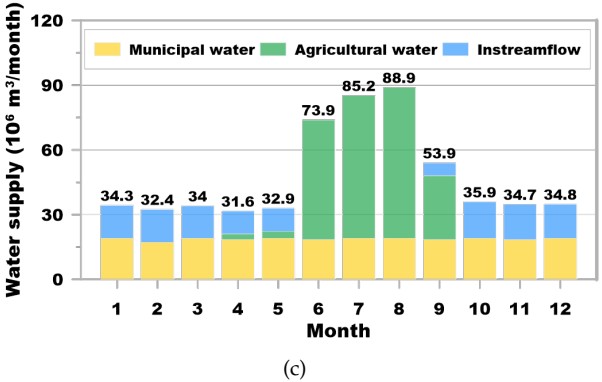

(c)

**Figure 6.** Monthly planned water supply: (**a**) Andong-Imha; (**b**) Hapcheon; (**c**) Namgang.

The reservoir project data and annual average inflow are presented in Table 2. The historical records for each reservoir, which are storage, inflow, and release in the reservoir, are shown in Figures A1–A3. The annual planned water supply of AD-IH is 1517 million m$^3$, which is 94% of the annual average inflow and 107% of the active capacity. The annual planned water supply of HC is 94% of the annual average inflow and 105% of the active capacity. AD-IH and HC, which are assigned much annual water supply compared to the average annual inflow, are bound to be vulnerable to droughts, and operation rules for water rationing are necessary. NG drains a relatively wide basin area compared to the capacity afford to supply, whereas the enforced water rationing was recorded from 2008 to 2009 due to an annual inflow of 799 million m$^3$ in 2008 (Figure A3). One of the meteorological characteristics of South Korea is that more than 70% of the annual precipitation is concentrated in the monsoon season (25 June to 25 September). For this reason, NG, which has an active capacity of 27% of the annual average inflow, spills most of the inflow during the monsoon season. The small active capacity of NG can be vulnerable to long-term drought, as NG has to release the annual planned water supply about three times the active capacity. For reference, NG may need to expand the dam to store more water, but it is challenging to expand a dam height due to topographical and environmental factors.

**Table 2.** Reservoir project data and annual average inflow.

| Reservoir | Watershed Area (km$^2$) | Annual Average Inflow (10$^6$ m$^3$) | Storage of Normal High Water Level (10$^6$ m$^3$) | Storage of Low-Water Level (10$^6$ m$^3$) | Active Capacity (10$^6$ m$^3$) |
|---|---|---|---|---|---|
| AD-IH | 2945 | 1613 | 1772 | 351.0 | 1421 |
| HC | 925.0 | 637.4 | 724.1 | 151.0 | 599.0 |
| NG | 2285 | 2105 | 182.4 | 16.15 | 166.3 |

Table 3 presents that the water rationing factors for each drought phase are calculated by applying the Korean water rationing strategy (2) to the planned water supply (6) of each reservoir. What is noteworthy in Table 3 is the water rationing factor for the severe phase of the NG from April to September Due to the drought in the NG from 2008 to 2009 (Figure A3), even the optimization results of applying the Korean water rationing strategy (agricultural water is released at the severe phase) could not avoid the failure of the water supply. Thus, the water rationing factor for the severe phase of the NG was calculated assuming that agricultural water was not released from April to September at the severe phase. While the reservoir capacity of NG is small, the annual average inflow is large, and the agricultural water supply is significantly dependent on the inflow of the monsoon season. Accordingly, in a severe drought, the supply of agricultural water may affect the supply of municipal water in the case of NG. Not supplying agricultural water at the severe phase may not be a reasonable option. The Korean water rationing strategy is also not appropriate as a water rationing strategy for NG to cope with drought. Therefore, an additional plan is needed for an appropriate water rationing strategy, considering the characteristics of the reservoir capacity, planned water supply, and inflow, rather than a uniform water rationing strategy in Korea.

**Table 3.** Rationing factor for monthly varying water rationing strategy.

| Month | Andong-Imha | | | | Hapcheon | | | | Namgang | | | |
|---|---|---|---|---|---|---|---|---|---|---|---|---|
| | $\alpha_{1,p}$ | $\alpha_{2,p}$ | $\alpha_{3,p}$ | $\alpha_{4,p}$ | $\alpha_{1,p}$ | $\alpha_{2,p}$ | $\alpha_{3,p}$ | $\alpha_{4,p}$ | $\alpha_{1,p}$ | $\alpha_{2,p}$ | $\alpha_{3,p}$ | $\alpha_{4,p}$ |
| January | 0.74 | 0.41 | 0.41 | 0.33 | 0.86 | 0.77 | 0.77 | 0.54 | 0.74 | 0.29 | 0.29 | 0.23 |
| February | 0.75 | 0.42 | 0.42 | 0.34 | 0.91 | 0.82 | 0.82 | 0.58 | 0.75 | 0.28 | 0.28 | 0.22 |
| March | 0.74 | 0.42 | 0.42 | 0.34 | 0.79 | 0.71 | 0.71 | 0.50 | 0.74 | 0.30 | 0.30 | 0.24 |
| April | 0.76 | 0.43 | 0.43 | 0.34 | 0.65 | 0.56 | 0.56 | 0.39 | 0.73 | 0.39 | 0.37 | 0.25 |
| May | 0.83 | 0.40 | 0.38 | 0.32 | 0.74 | 0.61 | 0.61 | 0.42 | 0.73 | 0.40 | 0.38 | 0.24 |
| June | 0.89 | 0.35 | 0.33 | 0.28 | 0.85 | 0.63 | 0.63 | 0.44 | 0.88 | 0.88 | 0.73 | 0.11 |
| July | 0.86 | 0.34 | 0.33 | 0.27 | 0.74 | 0.55 | 0.55 | 0.39 | 0.89 | 0.89 | 0.66 | 0.09 |
| August | 0.88 | 0.28 | 0.27 | 0.22 | 0.75 | 0.53 | 0.53 | 0.37 | 0.90 | 0.90 | 0.66 | 0.09 |
| September | 0.85 | 0.34 | 0.33 | 0.27 | 0.95 | 0.78 | 0.78 | 0.55 | 0.84 | 0.73 | 0.56 | 0.14 |
| October | 0.78 | 0.42 | 0.42 | 0.34 | 0.83 | 0.73 | 0.73 | 0.51 | 0.75 | 0.28 | 0.28 | 0.22 |
| November | 0.77 | 0.45 | 0.45 | 0.36 | 0.84 | 0.75 | 0.75 | 0.53 | 0.75 | 0.28 | 0.28 | 0.22 |
| December | 0.74 | 0.43 | 0.43 | 0.34 | 0.87 | 0.78 | 0.78 | 0.55 | 0.74 | 0.29 | 0.29 | 0.23 |

## 4. Results and Discussion

Applying the DDS and the DDS-FSR to the optimization models of section two yielded the hedging rule curve values, the parameters in the reservoir simulation models. Then, their performances were evaluated for practical optimization problems. The input data to derive hedging rule curves were monthly historical inflow records, planned water supply, and water rationing factors (Table 3) for each reservoir. The optimization period for AD-IH was from 1992 to 2020, for HC it was from 1989 to 2020, and for NG it was from 2002 to 2020 (Table 4). The objective function of the optimization problem is to minimize the water supply shortage within the optimization period. To verify the optimization results, the total water supply shortage for each reservoir was calculated based on the record data within the optimization period (Table 4). In the case of AD-IH, it can be shown in Figure A1 that three long-term droughts occurred in the optimization period, and the drought periods are as follows: August 1994 to February 1998, September 2006 to January 2010, and July 2013 to June 2016. HC was also rationed for a long time due to a long-term drought similar to AD-IH (Figure A2). In the case of AD-IH, it can be shown in Figure 3 that three long-term droughts occurred in the optimization period, and the drought periods are as follows: August 1994 to February 1998, September 2006 to January 2010, and July 2013 to June 2016. HC was also rationed for a long time due to the long-term drought similar to AD-IH. NG released less than 50% of the monthly total planned water supply due to difficulties in securing water from August 2008 to February 2009 (Figure A3). Accordingly, due to the

small inflow in the monsoon season in 2018, NG was preemptively rationed from December 2018 to June 2019 to avoid the same situation as in 2008.

**Table 4.** The historical records for the total water shortage and the optimization period to derive the hedging rule curves for each reservoir.

| Reservoir | Optimization Period | Total Water Supply Shortage ($10^6$ m$^3$) |
|:---:|:---:|:---:|
| AD-IH | January 1992~ December 2020 | 9481 |
| HC | January 1989~ December 2020 | 4562 |
| NG | January 2002~ December 2020 | 765 |

The neighborhood perturbation size (*r*) and the maximum number of function evaluations (*m*) also govern the performance of the DDS. The neighborhood perturbation size affected the range of decisions for candidate solutions, and the maximum number of function evaluations influenced optimization time and the quality of the derived best solution. Thus, the DDS and the DDS-FRS were evaluated under the conditions in Table 5.

**Table 5.** Summary of the test cases to compare performance between the DDS and DDS-FSR.

| Case | Maximum Number of Function Evaluations (*m*) | Neighborhood Perturbation size (*r*) |
|:---:|:---:|:---:|
| 1 | 100,000 | 0.2 |
| 2 | 100,000 | 0.1 |
| 3 | 50,000 | 0.2 |
| 4 | 50,000 | 0.1 |
| 5 | 20,000 | 0.2 |
| 6 | 20,000 | 0.1 |
| 7 | 10,000 | 0.2 |
| 8 | 10,000 | 0.1 |

The initial values of the decision variables for each case were the same values that made the penalty term zero, included in the objective function (Equation (3)). Table 6 shows the search boundary of the trigger volumes for the DDS-FSR and the DDS. It is not easy to define the search boundaries of the unknown trigger volumes because the trigger volumes in a subset are constrained by each other. Thus, in optimizing the hedging rule curves using the DDS, the search boundaries were defined as the storage of low-water level (S-LWL) and storage of normal high water level (S-NHWL). In the DDS-FSR, the upper bound of $V_{1,p}$ and the lower bound of $V_{4,p}$ are fixed search bounds as the user-defined data: S-NHWL and S-LWL, respectively. The others (e.g., the lower bound of $V_{1,p}$, the lower and upper bounds of $V_{2,p}$ and $V_{3,p}$, and the upper bound of $V_{4,p}$) can be flexibly changed to the best solution of the current iteration during the optimization process.

**Table 6.** The search boundary of decision variables to optimize the hedging rule curves from the DDS-FSR and the DDS: S-LWL is the storage of low-water level; S-NHWL is the storage of normal high water level; $V_{dp,p}^{best}$ is the best solution at the current iteration.

| Trigger Volume | DDS-FSR | | DDS | |
|:---:|:---:|:---:|:---:|:---:|
| | Lower Bound | Upper Bound | Lower Bound | Upper Bound |
| $V_{1,p}$ | $V_{2,p}^{best}$ | S-NHWL | | |
| $V_{2,p}$ | $V_{3,p}^{best}$ | $V_{1,p}^{best}$ | S-LWL | S-NHWL |
| $V_{3,p}$ | $V_{4,p}^{best}$ | $V_{2,p}^{best}$ | | |
| $V_{4,p}$ | S-LWL | $V_{3,p}^{best}$ | | |

The optimized objective function values revealed the performances of the two algorithms in Table 7. The two algorithms were performed with 10 trial optimizations for

each case and reservoir. Table 7 shows the mean, maximum (worst), minimum (best), and standard deviation (St. dev) of the objective function values. As shown in Table 7, for all cases, the two algorithms derived a significantly decreased total water supply shortage compared to the historical records in Table 4. However, comparing the performances of the two algorithms, the DDS-FSR has better values for most of the cases except for case 2 ($r$ = 0.1, $m$ = 100,000) in HC than those from the DDS. Specifically, in case 1 for NG, the DDS results converged on the global optimum value of 0 only five times in 10 trials, whereas the DDS-FSR results converged on 0 in all the trials. As seen in cases 1 and 2 with $m$ = 100,000, the improvements between the two methods were slight in the converged value of the objective function. The DDS-FSR, however, outperformed the DDS on cases 7 and 8 with a small maximum number of function evaluations of 10,000. On average, the optimized objective function values of the DDS-FSR improved by 11% in AD-IH, 4% in HC, and 33% in NG compared to the DDS in case 7 ($r$ = 0.2, $m$ = 10,000).

**Table 7.** Comparison of statistics of the converged objective function values between the DDS and DDS-FRS for the cases: the underlined value is superior to the other (unit: $10^6$ m$^3$).

| Case | Reservoir | DDS | | | | DDS-FSR | | | |
|------|-----------|------|------|-------|---------|------|------|-------|---------|
| | | Best | Mean | Worst | St. Dev | Best | Mean | Worst | St. Dev |
| 1 | AD-IH | 3831 | 3912 | 3996 | 43 | 3806 | 3874 | 4008 | 56 |
| | HC | 1983 | 2001 | 2058 | 21 | 1941 | 1972 | 1988 | 18 |
| | NG | 0 | 3 | 8 | 4 | 0 | 0 | 0 | 0 |
| 2 | AD-IH | 3814 | 3887 | 3964 | 48 | 3803 | 3859 | 3907 | 32 |
| | HC | 1948 | 1985 | 2047 | 24 | 1959 | 1995 | 2079 | 33 |
| | NG | 0 | 2 | 9 | 4 | 0 | 1 | 9 | 3 |
| 3 | AD-IH | 3935 | 4058 | 4345 | 111 | 3843 | 3885 | 3990 | 43 |
| | HC | 1984 | 2025 | 2192 | 59 | 1953 | 1987 | 2030 | 23 |
| | NG | 8 | 15 | 26 | 7 | 0 | 5 | 9 | 4 |
| 4 | AD-IH | 3876 | 3993 | 4202 | 91 | 3836 | 3891 | 3935 | 29 |
| | HC | 1955 | 2006 | 2111 | 42 | 1979 | 2000 | 2028 | 18 |
| | NG | 0 | 8 | 17 | 6 | 0 | 1 | 8 | 2 |
| 5 | AD-IH | 3948 | 4361 | 4867 | 330 | 3964 | 4132 | 4673 | 238 |
| | HC | 2014 | 2072 | 2155 | 44 | 1975 | 2027 | 2112 | 47 |
| | NG | 17 | 28 | 43 | 8 | 9 | 19 | 26 | 5 |
| 6 | AD-IH | 3978 | 4366 | 6072 | 583 | 3902 | 4088 | 4537 | 186 |
| | HC | 1983 | 2056 | 2196 | 60 | 1998 | 2045 | 2175 | 51 |
| | NG | 8 | 18 | 34 | 9 | 0 | 11 | 26 | 8 |
| 7 | AD-IH | 4337 | 4814 | 6510 | 590 | 4066 | 4301 | 4650 | 202 |
| | HC | 2016 | 2157 | 2451 | 126 | 1985 | 2067 | 2116 | 36 |
| | NG | 25 | 40 | 59 | 12 | 17 | 27 | 43 | 6 |
| 8 | AD-IH | 4193 | 4568 | 4977 | 254 | 4040 | 4450 | 4893 | 207 |
| | HC | 2022 | 2078 | 2190 | 59 | 2017 | 2073 | 2179 | 45 |
| | NG | 18 | 36 | 51 | 10 | 17 | 27 | 42 | 8 |

Figures 7–10 reveal some performances of the DDS and DDS-FSR by showing the converging processes of the objective function value. The plotted convergence processes the results from the 10 optimizations (gray line) and the mean values for each iteration (blue line). In Figures 7–10, the horizontal axis indicates the number of iterations, and the vertical axis indicates the current best value of the objective function. The DDS-FSR is faster than the DDS in convergence to an optimal function value. Specifically, in the convergence processes of the two methods for case 7 (Figures 9 and 10), the optimized results of the DDS-FSR are not only better, but also smaller in the deviations than the results from the DDS. Those results mean the following: the DDS-FSR can guide to a more promising search ranges as the continuously updated search boundaries; by minimizing overlapped search ranges, the DDS-FSR can avoid the candidate solutions that enlarge the penalty term of the objective function to imply the constraints. Furthermore, the DDS shows the difficulties of random functions and benchmark functions in the high-dimensional optimization problem

mentioned in Chu et al. [39,40] as an example in Figure 10. On the other hand, it can be seen in Figure 9 that the DDS-FSR has improved the difficulties of the DDS to some extent in the high-dimensional optimization problem by flexibly changing the search range.

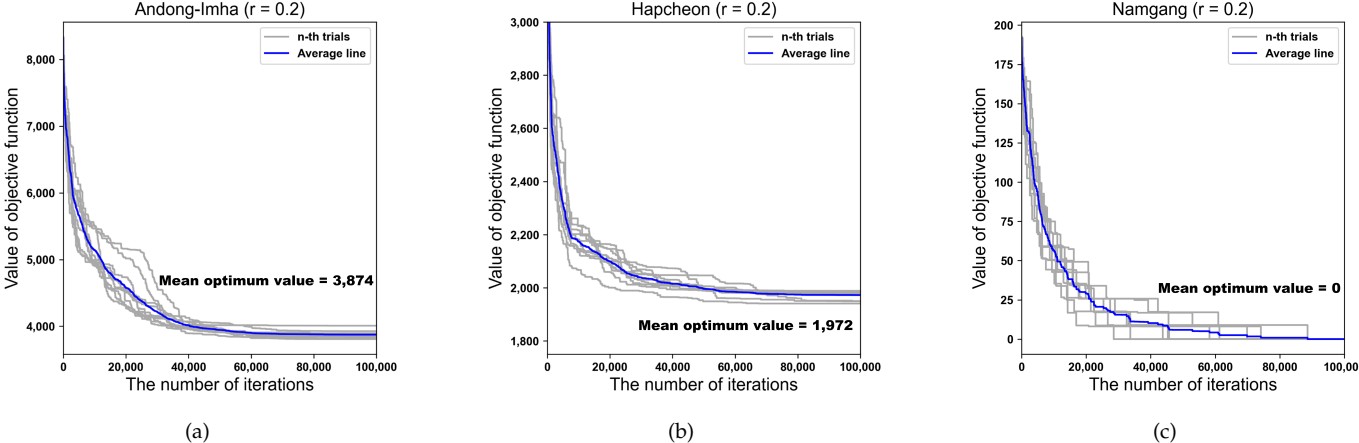

**Figure 7.** Convergence processes of the objective function value under the condition of case 1 (*r* = 0.2, *m* = 100,000) through 10 random optimization trials with the DDS-FRS: (**a**) Andong-Imha; (**b**) Hapcheon; (**c**) Namgang.

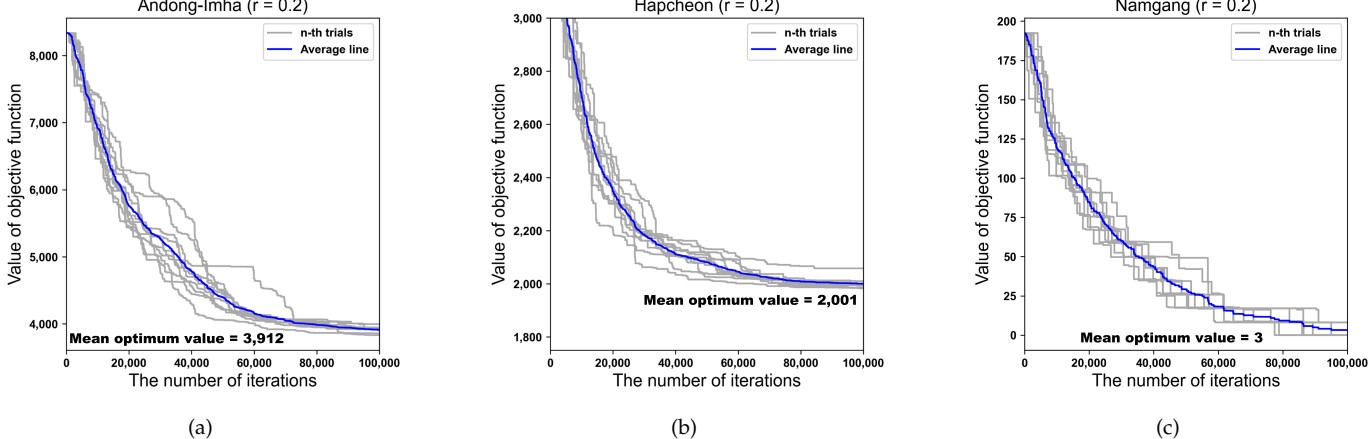

**Figure 8.** Convergence processes of the objective function value under the condition of case 1 (*r* = 0.2, *m* = 100,000) through 10 random optimization trials with the DDS: (**a**) Andong-Imha; (**b**) Hapcheon; (**c**) Namgang.

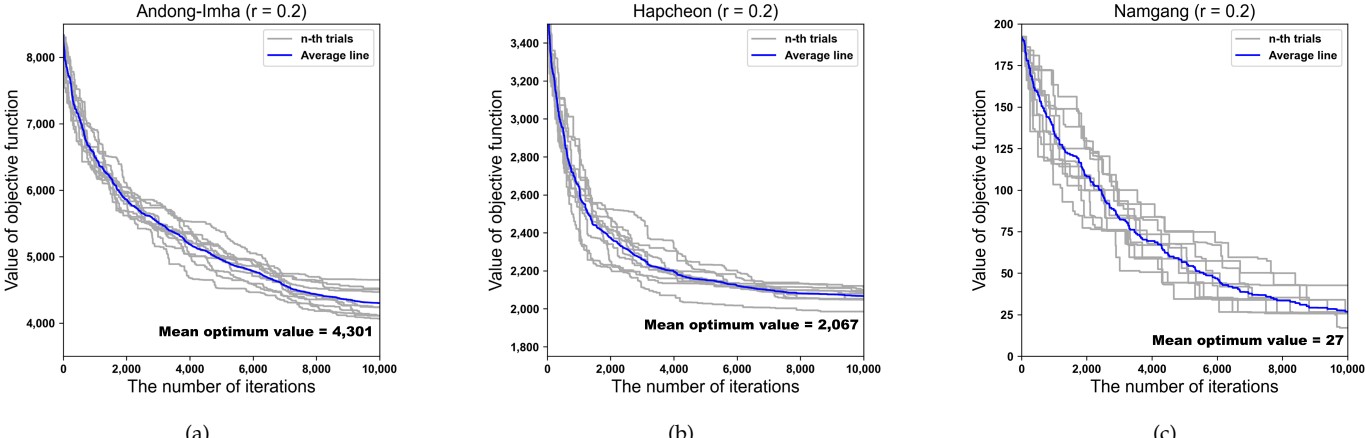

**Figure 9.** Convergence processes of the objective function value under the condition of case 7 ($r = 0.2$, $m = 10,000$) through 10 random optimization trials with the DDS-FRS: (**a**) Andong-Imha; (**b**) Hapcheon; (**c**) Namgang.

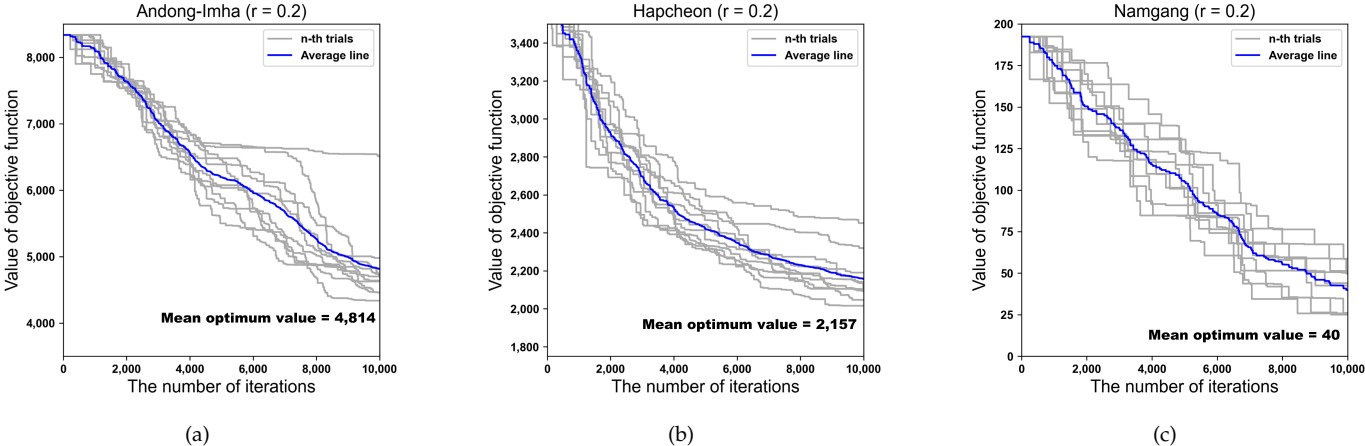

**Figure 10.** Convergence processes of the objective function value under the condition of case 7 ($r = 0.2$, $m = 10,000$) through 10 random optimization trials with the DDS: (**a**) Andong-Imha; (**b**) Hapcheon; (**c**) Namgang.

The unreasonable candidate solutions cause redundant iterations. Figures 11 and 12 present the candidate solutions for the two methods. Figure 12b shows an example of the unreasonable candidate solution in which the trigger volume of the severe drought phase (orange line) in December is greater than those of the concern and caution. In addition to the unreasonable candidate solutions presented in this paper (Figure 12b,d), a number of unreasonable candidate solutions were found in all the cases with DDS.

The DDS-FSR can reduce useless computations in function evaluations by minimizing the overlapped search range of decision variables. However, in cases 1 and 2, the optimization results of AD-IH and HC showed differences of about 1% between the DDS and the DDS-FSR. Those small differences were because the maximum number of function evaluations is so many that useless computation may be trivial.

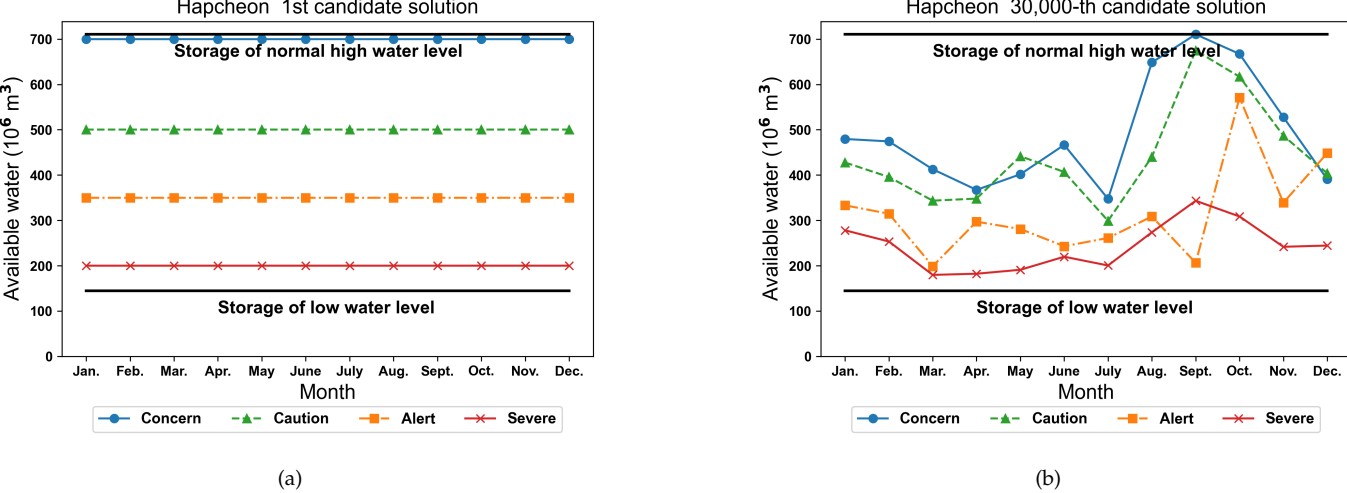

**Figure 11.** The attempted candidate solutions during the discrete hedging rule optimization for HC using the DDS-FSR, (**a**) the tried candidate solution at the initiation, (**b**) the tried candidate solution at the 30,000th, (**c**) the tried candidate solution at the 70,000th, (**d**) the tried candidate solution at the 100,000th.

**Figure 12.** *Cont.*

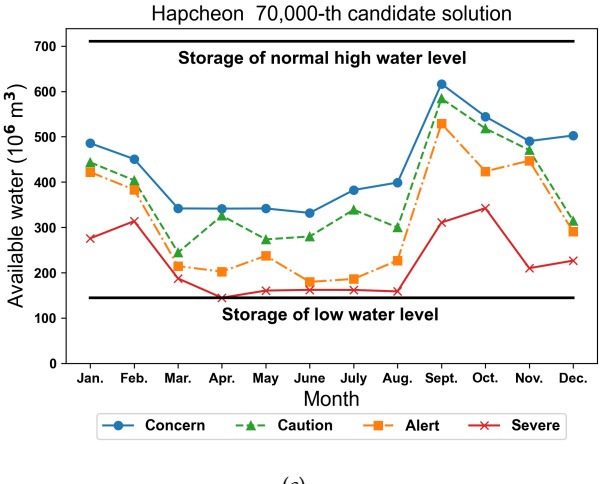

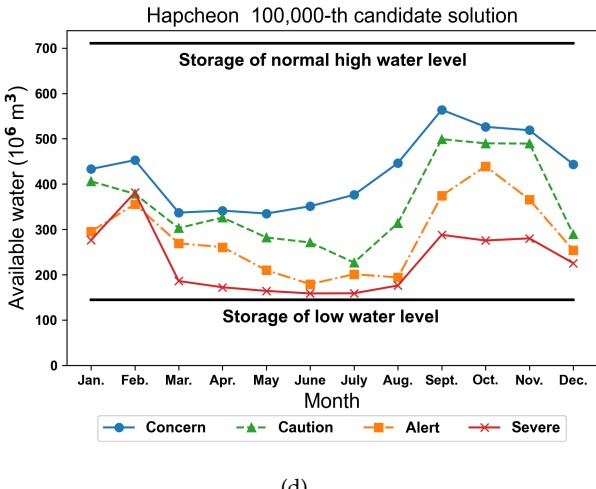

(c)

(d)

**Figure 12.** The attempted candidate solutions during the discrete hedging rule optimization for HC using the DDS, (**a**) the tried candidate solution at the initiation, (**b**) the tried candidate solution at the 30,000th, (**c**) the tried candidate solution at the 70,000th, (**d**) the tried candidate solution at the 100,000th.

The reservoir simulation results based on the hedging rule curves of HC, derived by the DDS-FSR in case 1 ($r$ = 0.2, $m$ = 100,000), are shown in Figures 13 and 14 for 2007 to 2020 among the simulation results from 1989 to 2020. In the reservoir simulation results from 1989 to 2020, the total water supply shortage was 2010 million m$^3$, which is 44% of the recorded total water supply shortage (4562 million m$^3$) at HC. In the simulation results from 2007 to 2020, shown in Figures 13 and 14, the total water supply shortage was 1056 million m$^3$, which is 46% of the recorded total water supply shortage (2283 million m$^3$) at HC in the same period. Thus, as suggested by previous studies related to hedging reservoir operation rules [12,13,15,23,26,43,48], it is analyzed that the hedging rule curves derived by DDS-FSR can also effectively cope with drought.

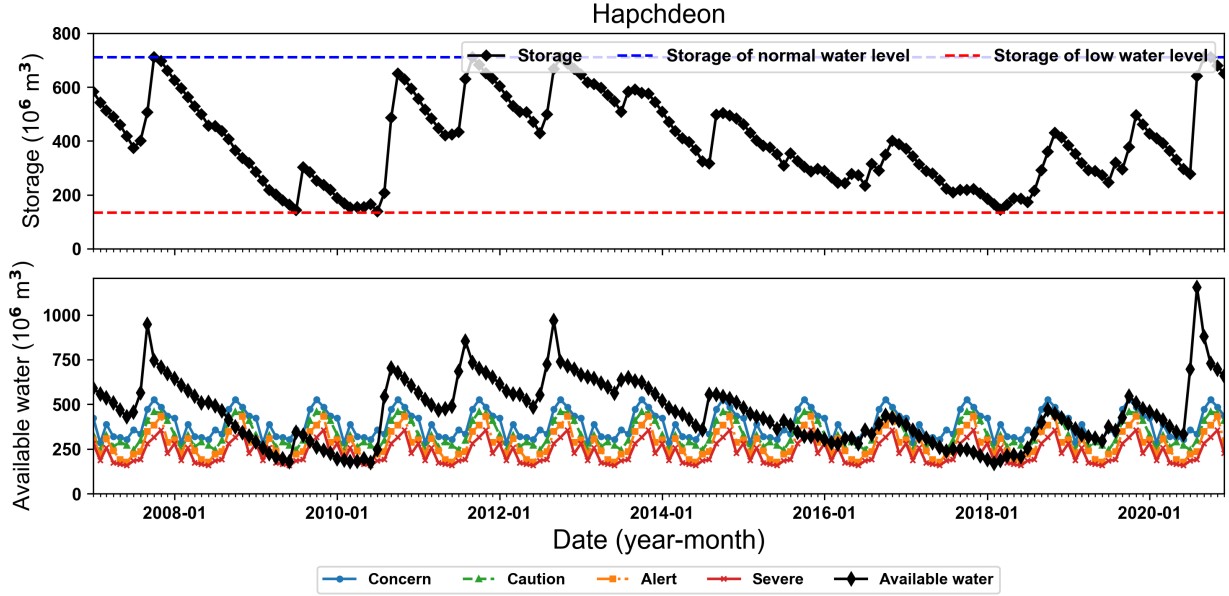

**Figure 13.** The reservoir simulation results for HC. The top plot shows the simulated monthly storage; the bottom plot shows the simulated monthly available water and the derived hedging rule curves for HC.

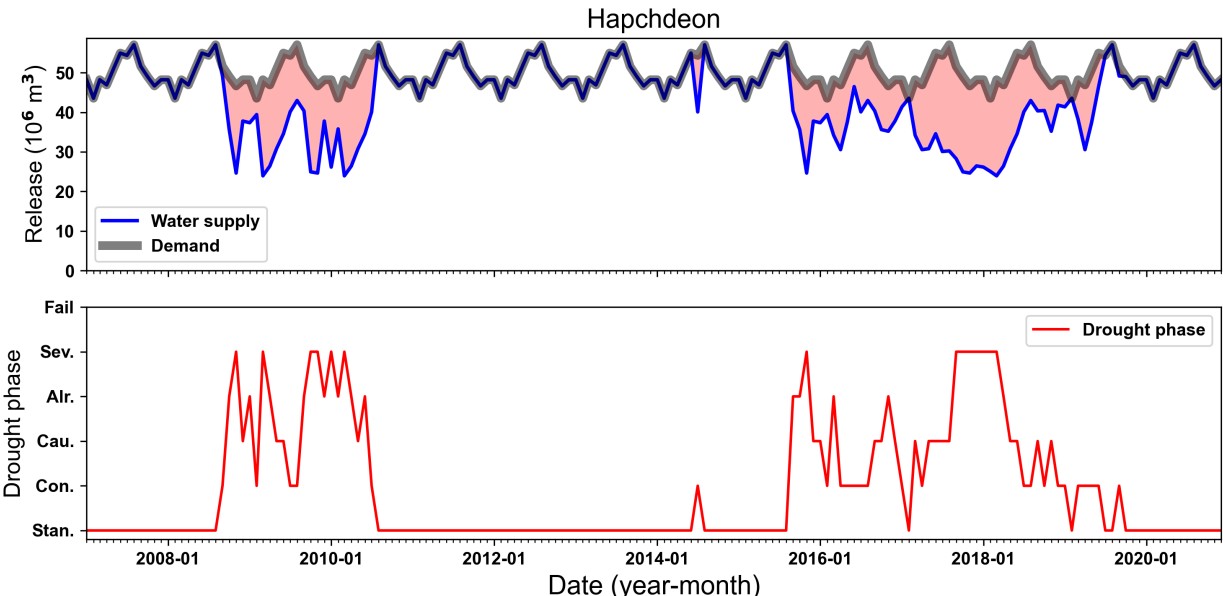

**Figure 14.** The reservoir simulation results for HC. The top plot shows the simulated monthly release and total planned water supply; the bottom shows the simulated monthly drought phase.

## 5. Conclusions

The discrete hedging rule can effectively manage water rationing from a reservoir to cope with droughts. The discrete hedging rule for reservoir operation includes time-varying trigger volumes used for the onset and termination of water rationing, which complicate its optimization problems by increasing the number of decision variables and constraints. The DDS can be easily applied to complex optimization problems, but its performance is relatively limited in constrained optimization problems such as deriving reservoir operation rules.

The DDS-FSR is proposed in this study to efficiently solve constrained optimization problems. The modified perturbing operator of the DDS-FSR can recursively update the search ranges of decision variables with limited overlaps.

The DDS and DDS-FSR were applied to derive the hedging rule curves for three reservoirs under three combinations of optimization parameters. The optimized objective function values and convergence processes revealed the performances of the two algorithms. The DDS-FSR closely converged to the optimum solutions at fewer evaluations than the DDS in all cases. Specifically, the smaller the maximum number of function evaluations was, the better the optimization performance of the DDS-FSR compared to the DDS. Those results mean the following: the DDS-FSR can guide to more promising search ranges as the continuously updated search boundaries; restraining the overlapped search ranges of decision variables, the DDS-FSR can avoid the candidate solutions that enlarge the penalty term of the objective function to imply the constraints. In other words, the DDS-FSR converged to a solution more efficiently and effectively than the DDS in the optimization problem for the hedging rule curves.

In practice, optimization problems might involve various constraints. The DDS-FSR was proposed as a simple strategy to optimize specific problems with the penalty term implying inequality constraint equations.

**Author Contributions:** Conceptualization, Y.J. and S.L.; methodology, Y.J.; software, Y.J.; validation, T.K., Y.J., and Y.K.; investigation, T.K.; data curation, T.K. and Y.J.; writing—original draft preparation, Y.J. and S.L.; writing—review and editing, Y.K. and S.L.; visualization, Y.J.; project administration, S.L. and Y.K. All authors have read and agreed to the submitted version of the manuscript.

**Funding:** This research was supported by Basic Science Research Program through the National Research Foundation of Korea (NRF), funded by the Ministry of Science, ICT & Future Planning (2017R1A2B2003715), and the Ministry of Education (NRF-2021R1F1A1062223). This work also was supported by the Korea Environmental Industry & Technology Institute (KEITI) through the Aquatic Ecosystem Conservation Research Program, funded by the Korea Ministry of Environment (MOE) (2020003050002).

**Data Availability Statement:** All the codes for the dynamically dimensioned search (DDS) algorithm and dynamically dimensioned search allowing a flexible search range (DDS-FSR) were written in Python (version 3.7) using PyCharm (https://www.jetbrains.com/pycharm/) accessed on 20 September 2022.The DDS is freely available via the link https://uwaterloo.ca/scholar/btolson/software-0 accessed on 20 September 2022. The DDS was developed by Bryon Tolson (btolson@uwaterloo.ca). The Python version of the DDS is freely available via GitHub, link https://github.com/t2abdulg/DDS_Py accessed on 20 September 2022. This version of the DDS was developed by Thouheed A.G. (https://github.com/t2abdulg). The DDS-FSR is freely available via GitHub, link https://github.com/LabWRSpknu/DDS_FSR_hedging_res accessed on 20 September 2022. The DDS-FSR was developed by Youngkyu Jin (accvn75@gmail.com) under the supervision of Sangho Lee (peterlee@pknu.ac.kr).

**Conflicts of Interest:** The authors declare no conflicts of interest.

## Appendix A. The Historical and Reservoir Operation Records for Andong-Imha, Hapcheon, and Namgang Reservoirs

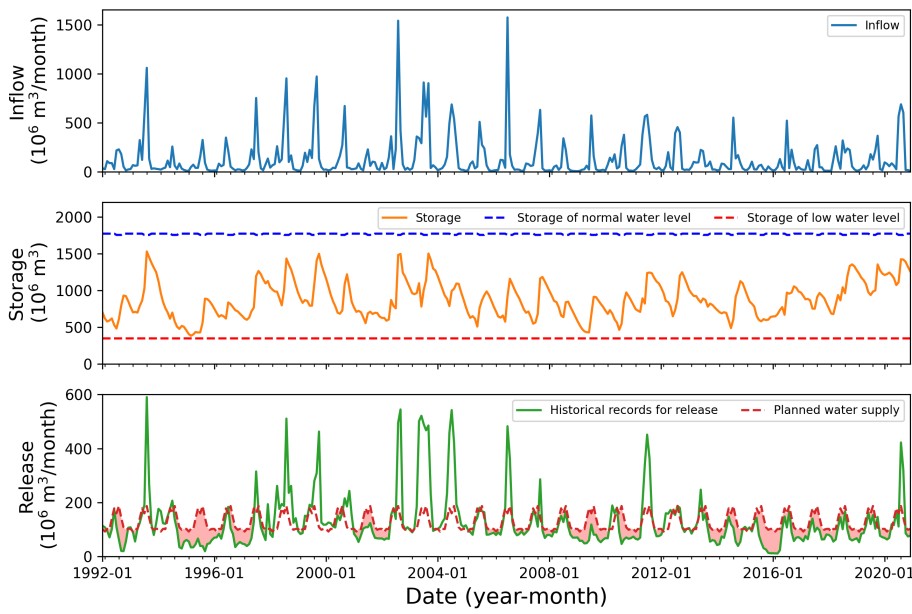

**Figure A1.** The historical records for Andong-Imha reservoir. The top plot shows the historical record of monthly inflow, the middle plot shows the reservoir operation records for storage, the bottom plot shows the monthly planned water supply and water supply records, and the areas filled with translucent red present the water supply shortage.

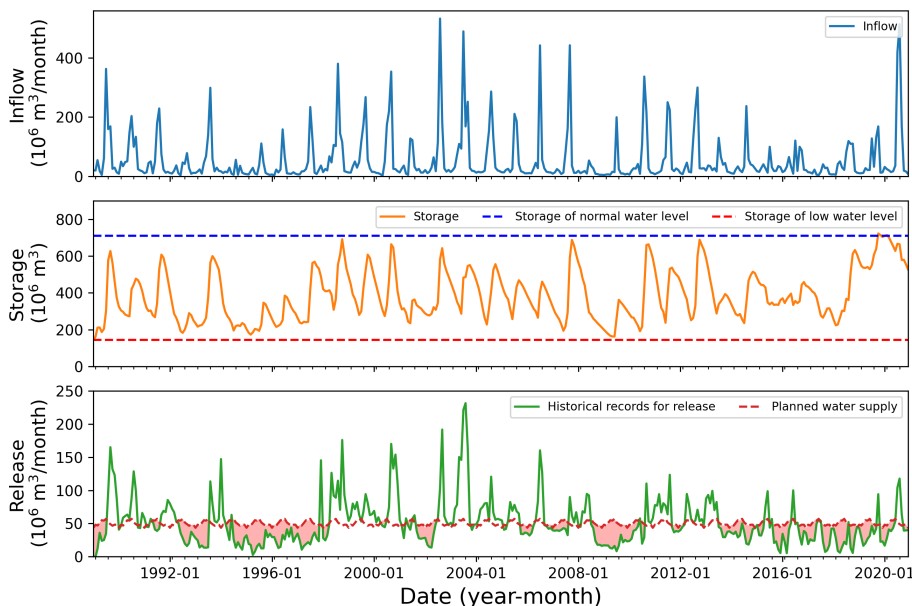

**Figure A2.** The historical records for Hapcheon reservoir. The top plot shows the historical record of monthly inflow, the middle plot shows reservoir operation records for storage, the bottom plot the monthly planned water supply and water supply records, and the areas filled with translucent red present the water supply shortage.

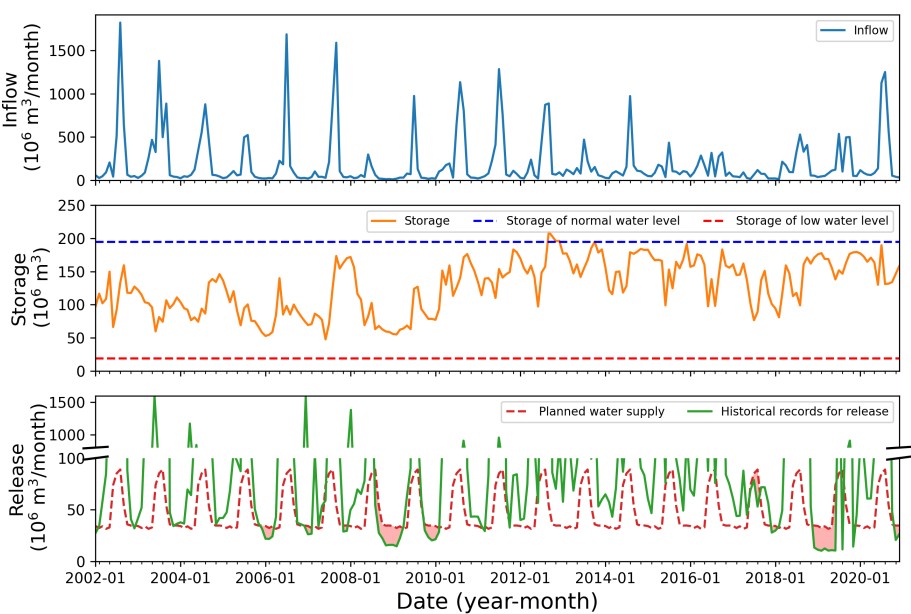

**Figure A3.** The historical records for Namgang reservoir. The top plot shows the historical record of monthly inflow, the middle plot shows reservoir operation records for storage, the bottom plot the monthly planned water supply and water supply records, and the areas filled with translucent red present the water supply shortage.

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
