# Peer review of "A Dynamically Dimensioned Search Allowing a Flexible Search Range and Its Application to Optimize Discrete Hedging Rule Curves"

_water, doi:10.3390/w14223633_

Round 1
Reviewer 1 Report
see below comments for editors.
The authors have done a lot of research and application work in optimization of discrete hedge rules, the research's method is correctly, data is detailed, the conclusion is correct.But there are a few small problems need to be modified. it will be listed as follows:
1. The author cannot do clear enough to express background description of the municipal water, agricultural water and instreamflow in three study work areas.
2. The authors need to remove notes of the Data Availability Statement in the paper as references in 3. Study area and data.
3. What's i and m meaning in Equation 2? What's alpha1 to alpha 4 meaning in Table 3?
4. The author needs to analyze what causes the water supply differences of municipal water, agricultural water and instreamflow in the three research area.
Author Response
We are grateful for your review and some meaningful comments.
We actively accepted the reviewer’s comments and revised most of the part of the paper.
The main revisions are as follows:
- We added diagrams and descriptions to compare the algorithms of the GA, SCE-UA, and DDS, which are representative heuristic algorithms.
- A detailed description of Korea's reduction supply strategy (especially how to calculate the water rationing factor) and planned water supply has been added.
- Historical records for the three reservoirs and analyzed results were added to verify the derived hedging rule curves by the optimization.
- We increased the number of test cases to compare the DDS and DDS-FSR from 3 to 8.
Please see the attachment for details.
Thank you for your consideration.

Reviewer 2 Report
1.The article, "Determination of hedging rule curves to mitigate water supply deficit for a single dam using dynamically dimensioned search method" by Jin, Y.; Lee, S. 11th World Congress on Water Resources and Environment: Managing Water Resources for a Sustainable Future - EWRA 2019. Proceedings" seems a little similar with your submitted. It may be a beginning try by DDS. The author should cite that as a reference.
2. The DDS-FSR is shown at the first time in line 175. There is no complete explain of the abbreviation of DDS-FSR before.
3. there are several gaps in the explanation of equations such as parameter “c " in Eq.(1), “m” and “n” in Eq.(2), parameter ‘r ‘in Line 130, … . And some variables in this study is not consistent.
4. Line 204~218, explain the project data, but I cannot find the input of the models. Which one, annual or monthly inflow, is input to the models? It’s not clear. It leads different model ability or need model validation.
5. Line 158, “the authors modified” is replaced by “this study”.
6. The study try to change the seed of search zone, which decrease the overly range, to make the optimization more efficient. From the results, the difference seems insignificant. One the other hand, lots of methods or modified processes can also make it efficient. For example, change the distribution of seed variables. Finally, the DDS-FSR a tiny issue on this case.
Author Response

(The authors gave the same response as above.)

Reviewer 3 Report
The authors have illustrated comprehensively the impact of using a DDS-FSR vs a DDS approach to establishing rules for reservoir storage. At present it is a theoretical discussion, that needs more context. In order for the utility of the DDS-FSR approach to be realized by most readers requires, comparison of a specific output in Figure 14 to actual management approach during a drought, how might this have helped in a specific previous drought?
3: Remove “its”
9: reword “optimum solutions with fewer evaluations using the modified algorithm than using the traditional algorithm.”
31: reword-“..optimization problems utilizing/generating hedging rules.”
38: reword “It is often impractical or unsuitable to apply linear and nonlinear programming to solve constrained optimization problems because of the amount of computation required 41
becomes unmanageable as the problem size increases;…”
63: Move this sentence to start of paragraph. “Some researchers have applied heuristic methods to 53
solve optimization problems in the water resources field due to flexibility and efficiency in 54
searching for optimum solutions [28–31].”
71: Can a diagram be used to illustrate the differing approaches of each heuristic method?
87: Reword “Programming can easily establish specific trigger volumes in the period..”
147: Explain penalty methods.
177: How are these three incorporated in either Figure 4 or Table 1.
1. The sum of the water supply shortage;
2. The penalty to restrain reversal of trigger volumes in drought phase severity;
3. The penalty to restrain water supply failures within the optimization period.
259: Indicate the units for Table 5.
Figure 7-11b. Reduce the y-axis value range to better differentiate the lines.
273: It is important to comment on the causes and significance of the better solutions with more solutions.
307: In practical terms how does the DDS-FSR tried candidate solution at the 100, 000th compare to the actual management observed during drought conditions?
310: Implying or informing or generating?
Author Response

(The authors gave the same response as above.)

Round 2
Reviewer 2 Report
Figure2 and Figure 5 could be combined into one.
Line 259, WDt is not matched with WSt in Eq.(9). (Typing error?)